# GRAPH WAVELET NEURAL NETWORK

**Bingbing Xu[1,2] , Huawei Shen[1,2], Qi Cao[1,2], Yunqi Qiu[1,2] & Xueqi Cheng[1,2]**
[1]CAS Key Laboratory of Network Data Science and Technology,
Institute of Computing Technology, Chinese Academy of Sciences;
[2]School of Computer and Control Engineering,
University of Chinese Academy of Sciences
Beijing, China
{xubingbing,shenhuawei,caoqi,qiuyunqi,cxq}@ict.ac.cn

## ABSTRACT

We present graph wavelet neural network (GWNN), a novel graph convolutional neural network (CNN), leveraging graph wavelet transform to address the shortcomings of previous spectral graph CNN methods that depend on graph Fourier transform. Different from graph Fourier transform, graph wavelet transform can be obtained via a fast algorithm without requiring matrix eigendecomposition with high computational cost. Moreover, graph wavelets are sparse and localized in vertex domain, offering high efficiency and good interpretability for graph convolution. The proposed GWNN significantly outperforms previous spectral graph CNNs in the task of graph-based semi-supervised classification on three benchmark datasets: Cora, Citeseer and Pubmed.

## 1 INTRODUCTION

Convolutional neural networks (CNNs) (LeCun et al., 1998) have been successfully used in many machine learning problems, such as image classification (He et al., 2016) and speech recognition (Hinton et al., 2012), where there is an underlying Euclidean structure. The success of CNNs lies in their ability to leverage the statistical properties of Euclidean data, e.g., translation invariance. However, in many research areas, data are naturally located in a non-Euclidean space, with graph or network being one typical case. The non-Euclidean nature of graph is the main obstacle or challenge when we attempt to generalize CNNs to graph. For example, convolution is not well defined in graph, due to that the size of neighborhood for each node varies dramatically (Bronstein et al., 2017).

Existing methods attempting to generalize CNNs to graph data fall into two categories, spatial methods and spectral methods, according to the way that convolution is defined. Spatial methods define convolution directly on the vertex domain, following the practice of the conventional CNN. For each vertex, convolution is defined as a weighted average function over all vertices located in its neighborhood, with the weighting function characterizing the influence exerting to the target vertex by its neighbors (Monti et al., 2017). The main challenge is to define a convolution operator that can handle neighborhood with different sizes and maintain the weight sharing property of CNN. Although spatial methods gain some initial success and offer us a flexible framework to generalize CNNs to graph, it is still elusive to determine appropriate neighborhood.

Spectral methods define convolution via graph Fourier transform and convolution theorem. Spectral methods leverage graph Fourier transform to convert signals defined in vertex domain into spectral domain, e.g., the space spanned by the eigenvectors of the graph Laplacian matrix, and then filter is defined in spectral domain, maintaining the weight sharing property of CNN. As the pioneering work of spectral methods, spectral CNN (Bruna et al., 2014) exploited graph data with the graph Fourier transform to implement convolution operator using convolution theorem. Some subsequent works make spectral methods spectrum-free (Defferrard et al., 2016; Kipf & Welling, 2017; Khasanova & Frossard, 2017), achieving locality in spatial domain and avoiding high computational cost of the eigendecomposition of Laplacian matrix.

In this paper, we present graph wavelet neural network to implement efficient convolution on graph data. We take graph wavelets instead of the eigenvectors of graph Laplacian as a set of bases, and define the convolution operator via wavelet transform and convolution theorem. Graph wavelet neural network distinguishes itself from spectral CNN by its three desirable properties: (1) Graph wavelets can be obtained via a fast algorithm without requiring the eigendecomposition of Laplacian matrix, and thus is efficient; (2) Graph wavelets are sparse, while eigenvectors of Laplacian matrix are dense. As a result, graph wavelet transform is much more efficient than graph Fourier transform; (3) Graph wavelets are localized in vertex domain, reflecting the information diffusion centered at each node (Tremblay & Borgnat, 2014). This property eases the understanding of graph convolution defined by graph wavelets.

We develop an efficient implementation of the proposed graph wavelet neural network. Convolution in conventional CNN learns an individual convolution kernel for each pair of input feature and output feature, causing a huge number of parameters especially when the number of features is high. We detach the feature transformation from convolution and learn a sole convolution kernel among all features, substantially reducing the number of parameters. Finally, we validate the effectiveness of the proposed graph wavelet neural network by applying it to graph-based semi-supervised classification. Experimental results demonstrate that our method consistently outperforms previous spectral CNNs on three benchmark datasets, i.e., Cora, Citeseer, and Pubmed.

## 2 OUR METHOD

### 2.1 PRELIMINARY

Let $\mathcal{G} = \{\mathbb{V}, \mathbb{E}, \boldsymbol{A}\}$ be an undirected graph, where $\mathbb{V}$ is the set of nodes with $|\mathbb{V}| = n$, $\mathbb{E}$ is the set of edges, and $\boldsymbol{A}$ is adjacency matrix with $A_{i,j} = A_{j,i}$ to define the connection between node $i$ and node $j$. The graph Laplacian matrix $\mathcal{L}$ is defined as $\mathcal{L} = \boldsymbol{D} - \boldsymbol{A}$ where $\boldsymbol{D}$ is a diagonal degree matrix with $D_{i,i} = \sum_j A_{i,j}$, and the normalized Laplacian matrix is $\boldsymbol{L} = \boldsymbol{I}_n - \boldsymbol{D}^{-1/2}\boldsymbol{A}\boldsymbol{D}^{-1/2}$ where $\boldsymbol{I}_n$ is the identity matrix. Since $\boldsymbol{L}$ is a real symmetric matrix, it has a complete set of orthonormal eigenvectors $\boldsymbol{U} = (\boldsymbol{u}_1, \boldsymbol{u}_2, ..., \boldsymbol{u}_n)$, known as Laplacian eigenvectors. These eigenvectors have associated real, non-negative eigenvalues $\{\lambda_l\}_{l=1}^n$, identified as the frequencies of graph. Eigenvectors associated with smaller eigenvalues carry slow varying signals, indicating that connected nodes share similar values. In contrast, eigenvectors associated with larger eigenvalues carry faster varying signals across connected nodes.

### 2.2 GRAPH FOURIER TRANSFORM

Taking the eigenvectors of normalized Laplacian matrix as a set of bases, graph Fourier transform of a signal $\boldsymbol{x} \in R^n$ on graph $\mathcal{G}$ is defined as $\hat{\boldsymbol{x}} = \boldsymbol{U}^\top \boldsymbol{x}$, and the inverse graph Fourier transform is $\boldsymbol{x} = \boldsymbol{U}\hat{\boldsymbol{x}}$ (Shuman et al., 2013). Graph Fourier transform, according to convolution theorem, offers us a way to define the graph convolution operator, denoted as $*_{\mathcal{G}}$. Denoting with $\boldsymbol{y}$ the convolution kernel, $*_{\mathcal{G}}$ is defined as

$$\boldsymbol{x} *_{\mathcal{G}} \boldsymbol{y} = \boldsymbol{U}\big((\boldsymbol{U}^\top \boldsymbol{y}) \odot (\boldsymbol{U}^\top \boldsymbol{x})\big), \tag{1}$$

where $\odot$ is the element-wise Hadamard product. Replacing the vector $\boldsymbol{U}^\top \boldsymbol{y}$ by a diagonal matrix $g_\theta$, then Hadamard product can be written in the form of matrix multiplication. Filtering the signal $x$ by the filter $g_\theta$, we can write Equation (1) as $\boldsymbol{U}g_\theta\boldsymbol{U}^\top \boldsymbol{x}$.

However, there are some limitations when using Fourier transform to implement graph convolution: (1) Eigendecomposition of Laplacian matrix to obtain Fourier basis $\boldsymbol{U}$ is of high computational cost with $O(n^3)$; (2) Graph Fourier transform is inefficient, since it involves the multiplication between a dense matrix $\boldsymbol{U}$ and the signal $\boldsymbol{x}$; (3) Graph convolution defined through Fourier transform is not localized in vertex domain, i.e., the influence to the signal on one node is not localized in its neighborhood. To address these limitations, ChebyNet (Defferrard et al., 2016) restricts convolution kernel $g_\theta$ to a polynomial expansion

$$g_\theta = \sum_{k=0}^{K-1} \theta_k \Lambda^k, \tag{2}$$

where $K$ is a hyper-parameter to determine the range of node neighborhoods via the shortest path distance, $\theta \in R^K$ is a vector of polynomial coefficients, and $\Lambda = \mathrm{diag}(\{\lambda_l\}_{l=1}^n)$. However, such a polynomial approximation limits the flexibility to define appropriate convolution on graph, i.e., with a smaller $K$, it's hard to approximate the diagonal matrix $g_\theta$ with $n$ free parameters. While with a larger $K$, locality is no longer guaranteed. Different from ChebyNet, we address the aforementioned three limitations through replacing graph Fourier transform with graph wavelet transform.

## 2.3 GRAPH WAVELET TRANSFORM

Similar to graph Fourier transform, graph wavelet transform projects graph signal from vertex domain into spectral domain. Graph wavelet transform employs a set of wavelets as bases, defined as $\psi_s = (\psi_{s1}, \psi_{s2}, ..., \psi_{sn})$, where each wavelet $\psi_{si}$ corresponds to a signal on graph diffused away from node $i$ and $s$ is a scaling parameter. Mathematically, $\psi_{si}$ can be written as

$$\psi_s = \boldsymbol{U} \boldsymbol{G}_s \boldsymbol{U}^\top, \tag{3}$$

where $\boldsymbol{U}$ is Laplacian eigenvectors, $\boldsymbol{G}_s = \mathrm{diag}(g(s\lambda_1), ..., g(s\lambda_n))$ is a scaling matrix and $g(s\lambda_i) = e^{\lambda_i s}$.

Using graph wavelets as bases, graph wavelet transform of a signal $\boldsymbol{x}$ on graph is defined as $\hat{\boldsymbol{x}} = \psi_s^{-1} x$ and the inverse graph wavelet transform is $x = \psi_s \hat{\boldsymbol{x}}$. Note that $\psi_s^{-1}$ can be obtained by simply replacing the $g(s\lambda_i)$ in $\psi_s$ with $g(-s\lambda_i)$ corresponding to a heat kernel (Donnat et al., 2018). Replacing the graph Fourier transform in Equation (1) with graph wavelet transform, we obtain the graph convolution as

$$\boldsymbol{x} *_\mathcal{G} \boldsymbol{y} = \psi_s((\psi_s^{-1}\boldsymbol{y}) \odot (\psi_s^{-1}\boldsymbol{x})). \tag{4}$$

Compared to graph Fourier transform, graph wavelet transform has the following benefits when being used to define graph convolution:

1. **High efficiency:** graph wavelets can be obtained via a fast algorithm without requiring the eigendecomposition of Laplacian matrix. In Hammond et al. (2011), a method is proposed to use Chebyshev polynomials to efficiently approximate $\psi_s$ and $\psi_s^{-1}$, with the computational complexity $O(m \times |\mathbb{E}|)$, where $|\mathbb{E}|$ is the number of edges and $m$ is the order of Chebyshev polynomials.

2. **High spareness:** the matrix $\psi_s$ and $\psi_s^{-1}$ are both sparse for real world networks, given that these networks are usually sparse. Therefore, graph wavelet transform is much more computationally efficient than graph Fourier transform. For example, in the Cora dataset, more than $97\%$ elements in $\psi_s^{-1}$ are zero while only less than $1\%$ elements in $\boldsymbol{U}^\top$ are zero (Table 4).

3. **Localized convolution:** each wavelet corresponds to a signal on graph diffused away from a centered node, highly localized in vertex domain. As a result, the graph convolution defined in Equation (4) is localized in vertex domain. We show the localization property of graph convolution in Appendix A. It is the localization property that explains why graph wavelet transform outperforms Fourier transform in defining graph convolution and the associated tasks like graph-based semi-supervised learning.

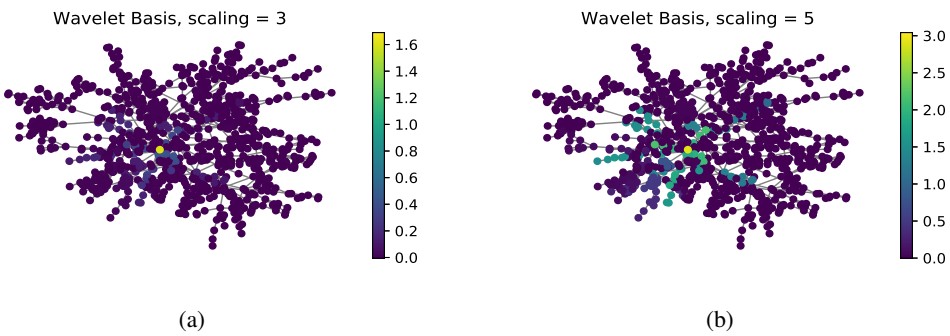

Figure 1: Wavelets on an example graph at (a) small scale and (b) large scale.

4. **Flexible neighborhood:** graph wavelets are more flexible to adjust node's neighborhoods. Different from previous methods which constrain neighborhoods by the discrete shortest path distance, our method leverages a continuous manner, i.e., varying the scaling parameter $s$. A small value of $s$ generally corresponds to a smaller neighborhood. Figure 1 shows two wavelet bases at different scale on an example network, depicted using GSP toolbox (Perraudin et al., 2014).

## 2.4 Graph wavelet neural network

Replacing Fourier transform with wavelet transform, graph wavelet neural network (GWNN) is a multi-layer convolutional neural network. The structure of the $m$-th layer is

$$\boldsymbol{X}_{[:,j]}^{m+1} = h(\psi_s \sum_{i=1}^{p} \boldsymbol{F}_{i,j}^{m} \psi_s^{-1} \boldsymbol{X}_{[:,i]}^{m}) \qquad j = 1, \cdots, q, \tag{5}$$

where $\psi_s$ is wavelet bases, $\psi_s^{-1}$ is the graph wavelet transform matrix at scale $s$ which projects signal in vertex domain into spectral domain, $\boldsymbol{X}_{[:,i]}^{m}$ with dimensions $n \times 1$ is the $i$-th column of $\boldsymbol{X}^m$, $\boldsymbol{F}_{i,j}^{m}$ is a diagonal filter matrix learned in spectral domain, and $h$ is a non-linear activation function. This layer transforms an input tensor $\boldsymbol{X}^m$ with dimensions $n \times p$ into an output tensor $\boldsymbol{X}^{m+1}$ with dimensions $n \times q$.

In this paper, we consider a two-layer GWNN for semi-supervised node classification on graph. The formulation of our model is

$$\text{first layer}: \ \boldsymbol{X}_{[:,j]}^{2} = \text{ReLU}(\psi_s \sum_{i=1}^{p} \boldsymbol{F}_{i,j}^{1} \psi_s^{-1} \boldsymbol{X}_{[:,i]}^{1}) \qquad j = 1, \cdots, q, \tag{6}$$

$$\text{second layer}: \ \boldsymbol{Z}_j = \text{softmax}(\psi_s \sum_{i=1}^{q} \boldsymbol{F}_{i,j}^{2} \psi_s^{-1} \boldsymbol{X}_{[:,,i]}^{2}) \qquad j = 1, \cdots, c, \tag{7}$$

where $c$ is the number of classes in node classification, $\boldsymbol{Z}$ of dimensions $n \times c$ is the prediction result. The loss function is the cross-entropy error over all labeled examples:

$$Loss = - \sum_{l \in y_L} \sum_{i=1}^{c} \boldsymbol{Y}_{li} \ln \boldsymbol{Z}_{li}, \tag{8}$$

where $y_L$ is the labeled node set, $\boldsymbol{Y}_{li} = 1$ if the label of node $l$ is $i$, and $\boldsymbol{Y}_{li} = 0$ otherwise. The weights $\boldsymbol{F}$ are trained using gradient descent.

## 2.5 Reducing Parameter Complexity

In Equation (5), the parameter complexity of each layer is $O(n \times p \times q)$, where $n$ is the number of nodes, $p$ is the number of features of each vertex in current layer, and $q$ is the number of features of each vertex in next layer. Conventional CNN methods learn convolution kernel for each pair of input feature and output feature. This results in a huge number of parameters and generally requires huge training data for parameter learning. This is prohibited for graph-based semi-supervised learning. To combat this issue, we detach the feature transformation from graph convolution. Each layer in GWNN is divided into two components: feature transformation and graph convolution. Specially, we have

$$\text{feature transformation}: \ \boldsymbol{X}^{m'} = \boldsymbol{X}^m \boldsymbol{W}, \tag{9}$$

$$\text{graph convolution}: \ \boldsymbol{X}^{m+1} = h(\psi_s \boldsymbol{F}^m \psi_s^{-1} \boldsymbol{X}^{m'}). \tag{10}$$

where $\boldsymbol{W} \in \mathbb{R}^{p \times q}$ is the parameter matrix for feature transformation, $\boldsymbol{X}^{m'}$ with dimensions $n \times q$ is the feature matrix after feature transformation, $\boldsymbol{F}^m$ is the diagonal matrix for graph convolution kernel, and $h$ is a non-linear activation function.

After detaching feature transformation from graph convolution, the parameter complexity is reduced from $O(n \times p \times q)$ to $O(n + p \times q)$. The reduction of parameters is particularly valuable fro graph-based semi-supervised learning where labels are quite limited.

# 3 RELATED WORKS

**Graph convolutional neural networks on graphs.** The success of CNNs when dealing with images, videos, and speeches motivates researchers to design graph convolutional neural network on graphs. The key of generalizing CNNs to graphs is defining convolution operator on graphs. Existing methods are classified into two categories, i.e., spectral methods and spatial methods.

Spectral methods define convolution via convolution theorem. Spectral CNN (Bruna et al., 2014) is the first attempt at implementing CNNs on graphs, leveraging graph Fourier transform and defining convolution kernel in spectral domain. Boscaini et al. (2015) developed a local spectral CNN approach based on the graph Windowed Fourier Transform. Defferrard et al. (2016) introduced a Chebyshev polynomial parametrization for spectral filter, offering us a fast localized spectral filtering method. Kipf & Welling (2017) provided a simplified version of ChebyNet, gaining success in graph-based semi-supervised learning task. Khasanova & Frossard (2017) represented images as signals on graph and learned their transformation invariant representations. They used Chebyshev approximations to implement graph convolution, avoiding matrix eigendecomposition. Levie et al. (2017) used rational functions instead of polynomials and created anisotropic spectral filters on manifolds.

Spatial methods define convolution as a weighted average function over neighborhood of target vertex. GraphSAGE takes one-hop neighbors as neighborhoods and defines the weighting function as various aggregators over neighborhood (Hamilton et al., 2017). Graph attention network (GAT) proposes to learn the weighting function via self-attention mechanism (Velickovic et al., 2017). MoNet offers us a general framework for design spatial methods, taking convolution as the weighted average of multiple weighting functions defined over neighborhood (Monti et al., 2017). Some works devote to making graph convolutional networks more powerful. Monti et al. (2018) alternated convolutions on vertices and edges, generalizing GAT and leading to better performance. GraphsGAN (Ding et al., 2018) generalizes GANs to graph, and generates fake samples in low-density areas between subgraphs to improve the performance on graph-based semi-supervised learning.

**Graph wavelets.** Sweldens (1998) presented a lifting scheme, a simple construction of wavelets that can be adapted to graphs without learning process. Hammond et al. (2011) proposed a method to construct wavelet transform on graphs. Moreover, they designed an efficient way to bypass the eigendecomposition of the Laplacian and approximated wavelets with Chebyshev polynomials. Tremblay & Borgnat (2014) leveraged graph wavelets for multi-scale community mining by modulating a scaling parameter. Owing to the property of describing information diffusion, Donnat et al. (2018) learned structural node embeddings via wavelets. All these works prove that graph wavelets are not only local and sparse but also valuable for signal processiong on graph.

# 4 EXPERIMENTS

## 4.1 DATASETS

To evaluate the proposed GWNN, we apply GWNN on semi-supervised node classification, and conduct experiments on three benchmark datasets, namely, Cora, Citeseer and Pubmed (Sen et al., 2008). In the three citation network datasets, nodes represent documents and edges are citation links. Details of these datasets are demonstrated in Table 1. Here, the label rate denotes the proportion of labeled nodes used for training. Following the experimental setup of GCN (Kipf & Welling, 2017), we fetch 20 labeled nodes per class in each dataset to train the model.

Table 1: The Statistics of Datasets

| Dataset | Nodes | Edges | Classes | Features | Label Rate |
|---|---|---|---|---|---|
| Cora | 2,708 | 5,429 | 7 | 1,433 | 0.052 |
| Citeseer | 3,327 | 4,732 | 6 | 3,703 | 0.036 |
| Pubmed | 19,717 | 44,338 | 3 | 500 | 0.003 |

## 4.2 BASELINES

We compare with several traditional semi-supervised learning methods, including label propagation (LP) (Zhu et al., 2003), semi-supervised embedding (SemiEmb) (Weston et al., 2012), manifold regularization (ManiReg) (Belkin et al., 2006), graph embeddings (DeepWalk) (Perozzi et al., 2014), iterative classification algorithm (ICA) (Lu & Getoor, 2003) and Planetoid (Yang et al., 2016).

Furthermore, along with the development of deep learning on graph, graph convolutional networks are proved to be effective in semi-supervised learning. Since our method is a spectral method based on convolution theorem, we compare it with the Spectral CNN (Bruna et al., 2014). ChebyNet (Defferrard et al., 2016) and GCN (Kipf & Welling, 2017), two variants of the Spectral CNN, are also included as our baselines. Considering spatial methods, we take MoNet (Monti et al., 2017) as our baseline, which also depends on Laplacian matrix.

## 4.3 EXPERIMENTAL SETTINGS

We train a two-layer graph wavelet neural network with 16 hidden units, and prediction accuracy is evaluated on a test set of 1000 labeled samples. The partition of datasets is the same as GCN (Kipf & Welling, 2017) with an additional validation set of 500 labeled samples to determine hyper-parameters.

Weights are initialized following Glorot & Bengio (2010). We adopt the Adam optimizer (Kingma & Ba, 2014) for parameter optimization with an initial learning rate $lr = 0.01$. For computational efficiency, we set the elements of $\psi_s$ and $\psi_s^{-1}$ smaller than a threshold $t$ to 0. We find the optimal hyper-parameters $s$ and $t$ through grid search, and the detailed discussion about the two hyper-parameters is introduced in Appendix B. For Cora, $s = 1.0$ and $t = 1e - 4$. For Citeseer, $s = 0.7$ and $t = 1e - 5$. For Pubmed, $s = 0.5$ and $t = 1e - 7$. To avoid overfitting, dropout (Srivastava et al., 2014) is applied. Meanwhile, we terminate the training if the validation loss does not decrease for 100 consecutive epochs.

## 4.4 ANALYSIS ON DETACHING FEATURE TRANSFORMATION FROM CONVOLUTION

Since the number of parameters for the undetached version of GWNN is $O(n \times p \times q)$, we can hardly implement this version in the case of networks with a large number $n$ of nodes and a huge number $p$ of input features. Here, we validate the effectiveness of detaching feature transformation form convolution on ChebyNet (introduced in Section 2.2), whose parameter complexity is $O(K \times p \times q)$. For ChebyNet of detaching feature transformation from graph convolution, the number of parameters is reduced to $O(K + p \times q)$. Table 2 shows the performance and the number of parameters on three datasets. Here, the reported performance is the optimal performance varying the order $K = 2, 3, 4$.

Table 2: Results of Detaching Feature Transformation from Convolution

|  | Method | Cora | Citeseer | Pubmed |
|---|---|---|---|---|
| **Prediction Accuracy** | ChebyNet | 81.2% | **69.8%** | 74.4% |
|  | Detaching-ChebyNet | **81.6%** | 68.5% | **78.6%** |
| **Number of Parameters** | ChebyNet | 46,080 (K=2) | 178,032 (K=3) | 24,144 (K=3) |
|  | Detaching-ChebyNet | 23,048 (K=4) | 59,348 (K=2) | 8,054 (K=3) |

As demonstrated in Table 2, with fewer parameters, we improve the accuracy on Pubmed by a large margin. This is due to that the label rate of Pubmed is only 0.003. By detaching feature transformation from convolution, the parameter complexity is significantly reduced, alleviating overfitting in semi-supervised learning and thus remarkably improving prediction accuracy. On Citeseer, there is a little drop on the accuracy. One possible explanation is that reducing the number of parameters may restrict the modeling capacity to some degree.

## 4.5 PERFORMANCE OF GWNN

We now validate the effectiveness of GWNN with detaching technique on node classification. Experimental results are reported in Table 3. GWNN improves the classification accuracy on all the three

Table 3: Results of Node Classification

| Method | Cora | Citeseer | Pubmed |
|---|---|---|---|
| MLP | 55.1% | 46.5% | 71.4% |
| ManiReg | 59.5% | 60.1% | 70.7% |
| SemiEmb | 59.0% | 59.6% | 71.7% |
| LP | 68.0% | 45.3% | 63.0% |
| DeepWalk | 67.2% | 43.2% | 65.3% |
| ICA | 75.1% | 69.1% | 73.9% |
| Planetoid | 75.7% | 64.7% | 77.2% |
| Spectral CNN | 73.3% | 58.9% | 73.9% |
| ChebyNet | 81.2% | 69.8% | 74.4% |
| GCN | 81.5% | 70.3% | 79.0% |
| MoNet | 81.7±0.5% | — | 78.8±0.3% |
| GWNN | **82.8%** | **71.7%** | **79.1%** |

datasets. In particular, replacing Fourier transform with wavelet transform, the proposed GWNN is comfortably ahead of Spectral CNN, achieving $10\%$ improvement on Cora and Citeseer, and 5% improvement on Pubmed. The large improvement could be explained from two perspectives: (1) Convolution in Spectral CNN is non-local in vertex domain, and thus the range of feature diffusion is not restricted to neighboring nodes; (2) The scaling parameter $s$ of wavelet transform is flexible to adjust the diffusion range to suit different applications and different networks. GWNN consistently outperforms ChebyNet, since it has enough degree of freedom to learn the convolution kernel, while ChebyNet is a kind of approximation with limited degree of freedom. Furthermore, our GWNN also performs better than GCN and MoNet, reflecting that it is promising to design appropriate bases for spectral methods to achieve good performance.

### 4.6 ANALYSIS ON SPARSITY

Besides the improvement on prediction accuracy, wavelet transform with localized and sparse transform matrix holds sparsity in both spatial domain and spectral domain. Here, we take Cora as an example to illustrate the sparsity of graph wavelet transform.

**The sparsity of transform matrix.** There are 2,708 nodes in Cora. Thus, the wavelet transform matrix $\psi_s^{-1}$ and the Fourier transform matrix $U^\top$ both belong to $\mathbb{R}^{2,708 \times 2,708}$. The first two rows in Table 4 demonstrate that $\psi_s^{-1}$ is much sparser than $U^\top$. Sparse wavelets not only accelerate the computation, but also well capture the neighboring topology centered at each node.

**The sparsity of projected signal.** As mentioned above, each node in Cora represents a document and has a sparse bag-of-words feature. The input feature $X \in \mathbb{R}^{n \times p}$ is a binary matrix, and $X_{[i,j]} = 1$ when the $i$-th document contains the $j$-th word in the bag of words, it equals 0 otherwise. Here, $X_{[:,j]}$ denotes the $j$-th column of $X$, and each column represents the feature vector of a word. Considering a specific signal $X_{[:,984]}$, we project the spatial signal into spectral domain, and get its projected vector. Here, $p = \psi_s^{-1} X_{[:,984]}$ denotes the projected vector via wavelet transform, $q = U^\top X_{[:,984]}$ denotes the projected vector via Fourier transform, and $p, q \in \mathbb{R}^{2,708}$. The last row in Table 4 lists the numbers of non-zero elements in $p$ and $q$. As shown in Table 4, with wavelet transform, the projected signal is much sparser.

Table 4: Statistics of wavelet transform and Fourier transform on Cora

| | Statistical Property | wavelet transform | Fourier transform |
|---|---|---|---|
| **Transform Matrix** | Density | 2.8% | 99.1% |
| | Number of Non-zero Elements | 205,774 | 7,274,383 |
| **Projected Signal** | Density | 10.9% | 100% |
| | Number of Non-zero Elements | 297 | 2,708 |

### 4.7 ANALYSIS ON INTERPRETABILITY

Compare with graph convolution network using Fourier transform, GWNN provides good interpretability. Here, we show the interpretability with specific examples in Cora.

Each feature, i.e. word in the bag of words, has a projected vector, and each element in this vector is associated with a spectral wavelet basis. Here, each basis is centered at a node, corresponding to a document. The value can be regarded as the relation between the word and the document. Thus, each value in $p$ can be interpreted as the relation between $Word_{984}$ and a document. In order to elaborate the interpretability of wavelet transform, we analyze the projected values of different feature as following.

Considering two features $Word_{984}$ and $Word_{1177}$, we select the top-10 active bases, which have the 10 largest projected values of each feature. As illustrated in Figure 2, for clarity, we magnify the local structure of corresponding nodes and marked them with bold rims. The central network in each subgraph denotes the dataset Cora, each node represents a document, and 7 different colors represent 7 classes. These nodes are clustered by OpenOrd (Martin et al., 2011) based on the adjacency matrix.

Figure 2a shows the top-10 active bases of $Word_{984}$. In Cora, this word only appears 8 times, and all the documents containing $Word_{984}$ belong to the class " Case-Based ". Consistently, all top-10 nodes activated by $Word_{984}$ are concentrated and belong to the class " Case-Based ". And, the frequencies of $Word_{1177}$ appearing in different classes are similar, indicating that $Word_{1177}$ is a universal word. In concordance with our expectation, the top-10 active bases of $Word_{1177}$ are discrete and belong to different classes in Figure 2b.

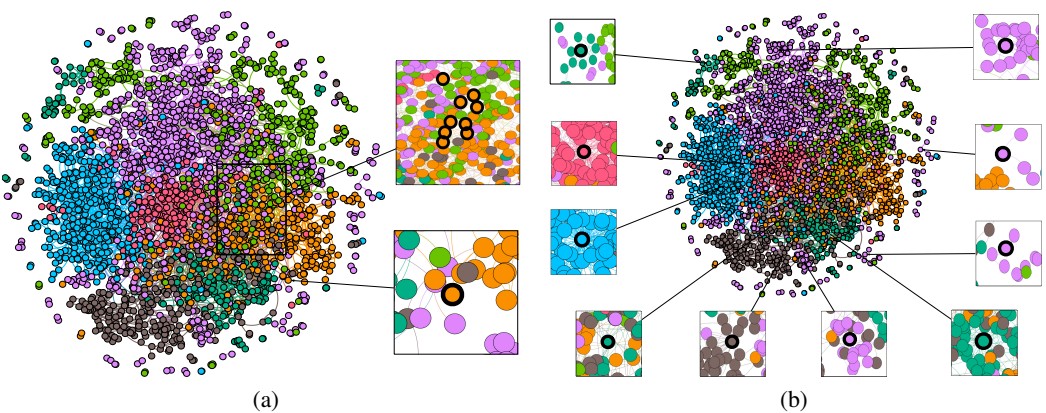

(a)                                                      (b)

Figure 2: Top-10 active bases of two words in Cora. The central network of each subgraph represents the dataset Cora, which is split into 7 classes. Each node represents a document, and its color indicates its label. The nodes that represent the top-10 active bases are marked with bold rims. (a) $Word_{984}$ only appears in documents of the class " Case-Based " in Cora. Consistently, all its 10 active bases also belong to the class " Case-Based ". (b) The frequencies of $Word_{1177}$ appearing in different classes are similar in Cora. As expected, the top-10 active bases of $Word_{1177}$ also belong to different classes.

Owing to the properties of graph wavelets, which describe the neighboring topology centered at each node, the projected values of wavelet transform can be explained as the correlation between features and nodes. These properties provide an interpretable domain transformation and ease the understanding of graph convolution.

## 5 CONCLUSION

Replacing graph Fourier transform with graph wavelet transform, we proposed GWNN. Graph wavelet transform has three desirable properties: (1) Graph wavelets are local and sparse; (2) Graph wavelet transform is computationally efficient; (3) Convolution is localized in vertex domain. These advantages make the whole learning process interpretable and efficient. Moreover, to reduce the number of parameters and the dependence on huge training data, we detached the feature transformation from convolution. This practice makes GWNN applicable to large graphs, with remarkable performance improvement on graph-based semi-supervised learning.

## 6 ACKNOWLEDGEMENTS

This work is funded by the National Natural Science Foundation of China under grant numbers 61425016, 61433014, and 91746301. Huawei Shen is also funded by K.C. Wong Education Foundation and the Youth Innovation Promotion Association of the Chinese Academy of Sciences.

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

## APPENDIX A    LOCALIZED GRAPH CONVOLUTION VIA WAVELET TRANSFORM

We use a diagonal matrix $\Theta$ to represent the learned kernel transformed by wavelets $\psi_s^{-1}\boldsymbol{y}$, and replace the Hadamard product with matrix muplication. Then Equation (4) is:

$$\boldsymbol{x} *_{\mathcal{G}} \boldsymbol{y} = \psi_s \Theta \psi_s^{-1} \boldsymbol{x}. \tag{11}$$

We set $\psi_s = (\psi_{s1}, \psi_{s2}, ..., \psi_{sn})$, $\psi_s^{-1} = (\psi_{s1}^*, \psi_{s2}^*, ..., \psi_{sn}^*)$, and $\Theta = \mathrm{diag}(\{\theta_k\}_{k=1}^n)$. Equation (11) becomes :

$$\boldsymbol{x} *_{\mathcal{G}} \boldsymbol{y} = \sum_{k=1}^n \theta_k \psi_{sk} (\psi_{sk}^*)^\top \boldsymbol{x}. \tag{12}$$

As proved by Hammond et al. (2011), both $\psi_s$ and $\psi_s^{-1}$ are local in small scale ($s$). Figure 3 shows the locality of $\psi_{s1}$ and $\psi_{s1}^*$, i.e., the first column in $\psi_s$ and $\psi_s^{-1}$ when $s = 3$. Each column in $\psi_s$ and $\psi_s^{-1}$ describes the neighboring topology of target node, which means that $\psi_s$ and $\psi_s^{-1}$ are local. The locality of $\psi_{sk}$ and $\psi_{sk}^*$ leads to the locality of the resulting matrix of multiplication between the column vector $\psi_{sk}$ and row vector $(\psi_{sk}^*)^\top$. For convenience, we set $\boldsymbol{M}_k = \psi_{sk}(\psi_{sk}^*)^\top$, $M_{k[i,j]} > 0$ only when $\psi_{sk}[i] > 0$ and $(\psi_{sk}^*)^\top[j] > 0$. In other words, if $\boldsymbol{M}_{k[i,j]} > 0$, vertex $i$ and vertex $j$ can correlate with each other through vertex $k$.

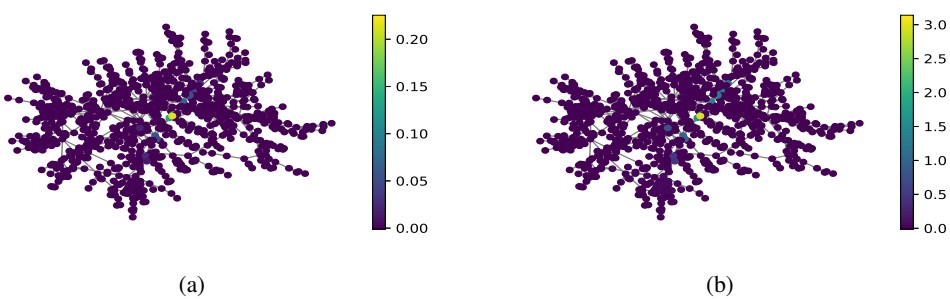

(a)                                                   (b)

Figure 3: Locality of (a) $\psi_{s1}$ and (b) $\psi_{s1}^*$.

Since each $\boldsymbol{M}_k$ is local, for any convolution kernel $\Theta$, $\psi_s \Theta \psi_s^{-1}$ is local, and it means that convolution is localized in vertex domain. By replacing $\Theta$ with an identity matrix in Equation (12), we get $\boldsymbol{x} *_{\mathcal{G}} \boldsymbol{y} = \sum_{k=1}^n \boldsymbol{M}_k \boldsymbol{x}$. We define $\boldsymbol{H} = \sum_{k=1}^n \boldsymbol{M}_k$, and Figure 4 shows $H_{[1,:]}$ in different scaling, i.e., correlation between the first node and other nodes during convolution. The locality of $\boldsymbol{H}$ suggests that graph convolution is localized in vertex domain. Moreover, as the scaling parameter $s$ becomes larger, the range of feature diffusion becomes larger.

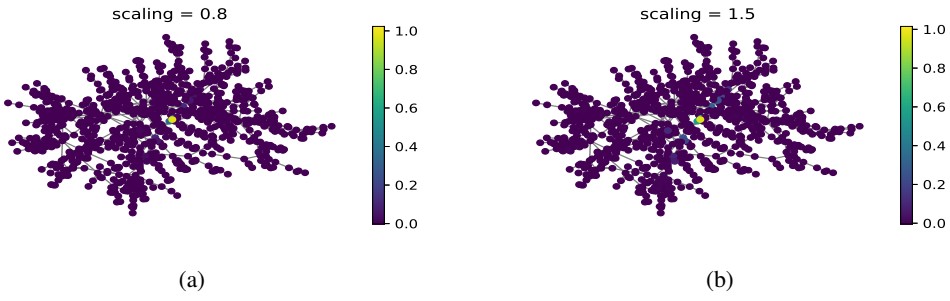

(a)                                                   (b)

Figure 4: Correlation between first node and other nodes at (a) small scale and (b) large scale. Non-zero value of node represents correlation between this node and target node during convolution. Locality of $\boldsymbol{H}$ suggests that graph convolution is localized in vertex domain. Moreover, with scaling parameter $s$ becoming larger, the range of feature diffusion becomes larger.

## APPENDIX B    INFLUENCE OF HYPER-PARAMETERS

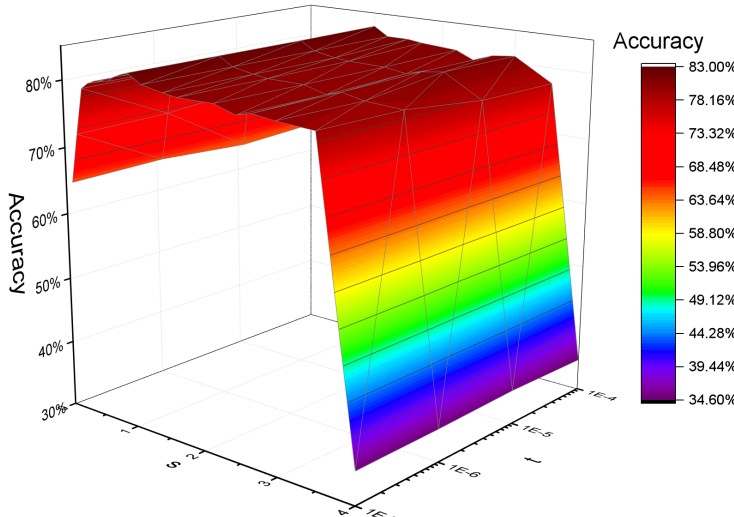

Figure 5: Influence of $s$ and $t$ on Cora.

GWNN leverages graph wavelets to implement graph convolution, where $s$ is used to modulate the range of neighborhoods. From Figure 5, as $s$ becomes larger starting from 0, the range of neighboring nodes becomes large, resulting the increase of accuracy on Cora. However when $s$ becomes too large, some irrelevant nodes are included, leading to decreasing of accuracy. The hyper-parameter $t$ only used for computational efficiency, has any slight influence on its performance.

For experiments on specific dataset, $s$ and $t$ are choosen via grid search using validation. Generally, a appropriate $s$ is in the range of $[0.5, 1]$, which can not only capture the graph structure but also guarantee the locality of convolution, and $t$ is less insensive to dataset.

## APPENDIX C    PARAMETER COMPLEXITY OF NODE CLASSIFICATION

We show the parameter complexity of node classification in Table 5. The high parameter complexity $O(n * p * q)$ of Spectral CNN makes it difficult to generalize to real world networks. ChebyNet approximates the convolution kernel via polynomial function of the diagonal matrix of Laplacian eigenvalues, reducing parameter complexity to $O(K * p * q)$ with $K$ being the order of polynomial function. GCN simplifies ChebyNet via setting $K$=1. We detach feature transformation from graph convolution to implement GWNN and Spectral CNN in our experiments, which can reduce parameter to $O(n + p * q)$.

Table 5: Parameter complexity of Node Classification

| Method | Cora | Citeseer | Pubmed |
|---|---|---|---|
| Spectral CNN | 62,392,320 | 197,437,488 | 158,682,416 |
| Spectral CNN (detaching) | 28,456 | 65,379 | 47,482 |
| ChebyNet | 46,080 (K=2) | 178,032 (K=3) | 24,144 (K=3) |
| GCN | 23,040 | 59,344 | 8,048 |
| GWNN | 28,456 | 65,379 | 47,482 |

In Cora and Citeseer, with smaller parameter complexity, GWNN achieves better performance than ChebyNet, reflecting that it is promising to implement convolution via graph wavelet transform. As Pubmed has a large number of nodes, the parameter complexity of GWNN is larger than ChebyNet. As future work, it is an interesting attempt to select wavelets associated with a subset of nodes, further reducing parameter complexity with potential loss of performance.

## APPENDIX D   FAST GRAPH WAVELETS WITH CHEBYSHEV POLYNOMIAL APPROXIMATION

Hammond et al. (2011) proposed a method, using Chebyshev polynomials to efficiently approximate $\psi_s$ and $\psi_s^{-1}$. The computational complexity is $O(m \times |\mathbb{E}|)$, where $|\mathbb{E}|$ is the number of edges and $m$ is the order of Chebyshev polynomials. We give the details of the approximation proposed in Hammond et al. (2011).

With the stable recurrence relation $T_k(y) = 2yT_{k-1}(y) - T_{k-2}(y)$, we can generate the Chebyshev polynomials $T_k(y)$. Here $T_0 = 1$ and $T_1 = y$. For $y$ sampled between -1 and 1, the trigonometric expression $T_k(y) = cos(k\,arccos(y))$ is satisfied. It shows that $T_k(y) \in [-1, 1]$ when $y \in [-1, 1]$. Through the Chebyshev polynomials, an orthogonal basis for the Hilbert space of square integrable functions $L^2([-1, 1], \frac{dy}{\sqrt{1-y^2}})$ is formed. For each $h$ in this Hilbert space, we have a uniformly convergent Chebyshev series $h(y) = \frac{1}{2}c_0 + \sum_{k=1}^{\infty} c_k T_k(y)$, and the Chebyshev coefficients $c_k = \frac{2}{\pi} \int_{-1}^{1} \frac{T_k(y)h(y)}{\sqrt{1-y^2}} dy = \frac{2}{\pi} \int_0^{\pi} cos(k\theta)h(cos(\theta))d\theta$. A fixed scale $s$ is assumed. To approximate $g(sx)$ for $x \in [0, \lambda_{max}]$, we can shift the domain through the transformation $x = a(y + 1)$, where $a = \frac{\lambda_{max}}{2}$. $T_k'(x) = T_k(\frac{x-a}{a})$ denotes the shifted Chebyshev polynomials, with $\frac{x-a}{a} \in [-1, 1]$. Then we have $g(sx) = \frac{1}{2}c_0 + \sum_{k=1}^{\infty} c_k T_k'(x)$, and $x \in [0, \lambda_{max}]$, $c_k = \frac{2}{\pi} \int_0^{\pi} cos(k\theta)g(s(a(cos(\theta) + 1)))d\theta$. we truncate the Chebyshev expansion to $m$ terms and achieve Polynomial approximation.

Here we give the example of the $\psi_s^{-1}$ and $g(sx) = e^{-sx}$, the graph signal is $\boldsymbol{f} \in R^n$. Then we can give the fast approximation wavelets by $\psi_s^{-1}\boldsymbol{f} = \frac{1}{2}c_0\boldsymbol{f} + \sum_{k=1}^{m} c_k T_k'(\boldsymbol{L})\boldsymbol{f}$. The efficient computation of $T_k'(\boldsymbol{L})$ determines the utility of this approach, where $T_k'(\boldsymbol{L})\boldsymbol{f} = \frac{2}{a}(\boldsymbol{L} - \boldsymbol{I})(T_{k-1}'(\boldsymbol{L})\boldsymbol{f}) - T_{k-2}'(\boldsymbol{L})\boldsymbol{f}$.

## APPENDIX E   ANALYSIS ON SPASITY OF SPECTRAL TRANSFORM AND LAPLACIAN MATRIX

The sparsity of the graph wavelets depends on the sparsity of the Laplacian matrix and the hyper-parameter $s$, We show the sparsity of spectral transform matrix and Laplacian matrix in Table 6.

Table 6: Statistics of spectral transform and Laplacian matrix on Cora

|  | Density | Num of Non-zero Elements |
|---|---|---|
| **wavelet transform** | 2.8% | 205,774 |
| **Fourier transform** | 99.1% | 7,274,383 |
| **Laplacian matrix** | 0.15% | 10,858 |

The sparsity of Laplacian matrix is sparser than graph wavelets, and this property limits our method, i.e., the higher time complexity than some methods depending on Laplacian matrix and identity matrix, e.g., GCN. Specifically, both our method and GCN aim to improve Spectral CNN via designing localized graph convolution. GCN, as a simplified version of ChebyNet, leverages Laplacian matrix as weighted matrix and expresses the spectral graph convolution in spatial domain, acting as spatial-like method (Monti et al., 2017). However, our method resorts to using graph wavelets as a new set of bases, directly designing localized spectral graph convolution. GWNN offers a localized graph convolution via replacing graph Fourier transform with graph wavelet transform, finding good spectral basis with localization property and good interpretability. This distinguishes GWNN from ChebyNet and GCN, which express the graph convolution defined via graph Fourier transform in vertex domain.

