# OpenReview forum: "Graph Wavelet Neural Network"
_ICLR.cc/2019/Conference_

### Official Review · AnonReviewer1 · 2018-11-01
**Interesting approach**

**Rating:** 7
**Confidence:** 4

**Review:**

This is an empirical paper that proposes to design wavelets on graphs, that can be integrated to neural networks on graphs. It permits to reducing the number of parameters of the « convolution » and exploits the sparsity of sparse weighted graphs for computations. I think it’s an interesting work.

The perspective I enjoy in using wavelets is that they typically provide a good trade-off in localization in the spectral and graph domains. For instance, large eigenvalues of the laplacian could be potentially captured in a more stable way. This type of work might be a first step.

Pros :
- good numerical results
- nice incorporation of structure via wavelets
Cons :
- Sometimes the paper is not really clear

I have severals comments:

1/ (1) « Some subsequent works devote to making spectral methods spectrum-free… avoiding high computational cost ». I think also those types of representations can be potentially unstable. That’s a second reason.

2/ I’m not sure to understand the point (1) of the fourth paragraph of the introduction. (1.)« graph wavelet does not depend on the eigen decomposition of Laplacian matrix ». Does it mean numerically ? This sentence is not clear, because even numerically, it can be done in a dependent way to this matrix. However, if it is implied that the fastest algorithm can be obtained without eigen-decomposition of the Laplacian Matrix, in a cheap way, then I agree and a small rephrasing could be nice.

3/ Please remove all the sentences that are supposed to be an introduction for a section, i.e. the sentences between 2 and 2.1 (« We use graph… ») and 3 and 3.1 (« In many research… »). They are poorly written and do not help the reader.

5/ .(2.2) “However, such a polynomial approximation also limits the flexibility to define appropriate convolution on graph” I’m not sure to understand. In the paper you refer to, the set of filters span the polynomial of degree less than n, of a diagonalizable matrix of size nxn. Thus, lagrange polynomial basis could be used to interpolate any desired values? Does it signify learning non-diagonal(in the appropriate basis) matrix?

6/ (2.3) $s$ how is this parameter chosen, crossvalidation? Is it adjusted such that the singular values of $\psi_s$ have a certain decay? Why is $s$ constant across layers? How is the spectrum of psi_s? Is it well conditionned? A huge difference with (Hammond et al,2011) is that they use a collection of wavelets. Did you consider this kind of approach ? Is there a loss of information in some cases?(like if $\psi_s$ has a fast decay)

7/ (2.3) “The matrix $\psi_s$ and $\psi_s^{-1}$ are both sparse”. This is a critical affirmation which is not in general true. It is possibly sparse if the weighted laplacian graph is sparse as well, as explained by the remark after theorem 5.5 of page 16 of (Hammond et al, 2011). However, I do agree this typically happens in the application you present.

8/ The (c) of Figure 1 is missing.(in my version at least)

9/ In section 5.3, why not trying to compare the number of parameters with other papers? I think also more results, with maybe a bit higher performances, could be reported, such as: GraphSGAN, GAT. But that’s fine to me.

10/ Appendix A, isn’t it a simple rephrasing of (Hammond et al, 2011)?

11/ How do you compare in term of interpretability with [1]?

12/ Just as a suggestion and/or a comment: it seems similar to approaches such as lifting schemes(which basically builds wavelets on graphs/manifolds), except that there is no learning involved. (e.g. [2]) I think there could be great connexions.


[1] Relational inductive biases, deep learning, and graph networks, Battaglia et al 2011?
[2] THE LIFTING SCHEME: A CONSTRUCTION OF SECOND GENERATION WAVELETS, Wim Sweldgens

---

> ### Author Response · Authors · 2018-11-19
> **Response to Reviewer1 (2/2)**
>
> Q7: (2.3) “The matrix $\psi_s$ and $\psi_s^{-1}$ are both sparse”. This is a critical affirmation which is not in general true. It is possibly sparse if the weighted laplacian graph is sparse as well, as explained by the remark after theorem 5.5 of page 16 of (Hammond et al, 2011). However, I do agree this typically happens in the application you present.
>
> A7: Thank you for drawing our attention to this point. Indeed, the sparsity of the matrix $\psi_s$ and $\psi_s^{-1}$ is related to the sparsity of the Laplacian matrix. Generally, real world networks are sparse with the number of edges much less than the square of the number of nodes. In these cases, the matrix $\psi_s$ and $\psi_s^{-1}$ are both sparse. Prompted by your comments, we revised the statement to offer an accurate claim.
>
> Q8: The (c) of Figure 1 is missing.(in my version at least)
>
> A8: We corrected the caption of Figure 1 in the revised version.
>
> Q9: In section 5.3, why not trying to compare the number of parameters with other papers? I think also more results, with maybe a bit higher performances, could be reported, such as: GraphSGAN, GAT. But that’s fine to me.
>
> A9: We agree that it is always better to compare with more methods. Yet, the major contribution is about improving spectral methods by using graph wavelets instead of eigenvectors as a set of bases. Therefore, we focus on the comparison with spectral methods. We particularly appreciate your understanding.
>
> Q10: Appendix A, isn’t it a simple rephrasing of (Hammond et al, 2011)?
>
> A10: Hammond et al. (2011) described the locality of graph wavelets. Instead, in Appendix A, we demonstrate that the graph convolution via graph wavelets is localized, i.e., the nodes used to update target node are its neighboring nodes. We clarified this point in the revised version.
>
> Q11: How do you compare in term of interpretability with [1]?
>
> A11: We agree that the meaning of “interpretability” is diverse in the literature. In this paper, we use “interpretability” to offer some intuitive understanding for the wavelet transform, compared with the Fourier transform. In [1], the proposed graph neural network, defining a flexible network via graph block, offers the interpretability as the correlation among nodes in spatial domain.
>
> Q12: Just as a suggestion and/or a comment: it seems similar to approaches such as lifting schemes (which basically builds wavelets on graphs/manifolds), except that there is no learning involved. (e.g. [2]) I think there could be great connexions.
>
> A12: The paper you mentioned presents a simple construction of wavelets that could be adapted to graphs without learning process. We cited this paper as related work in the revised version.

---

> > ### Comment · AnonReviewer1 · 2018-11-22
> > **Thanks for the clarifications**
> >
> > Dear authors,
> >
> > Thank you for your clarifications and revisions. Maybe one question related to Q9: do you have a rough idea of the number of parameters w.r.t. other methods? (I do not think I saw this in the revised version of the paper)
> >
> > I'd be happy to raise my score to 7 after this.
> > Best,

---

> > > ### Author Response · Authors · 2018-11-24
> > > **Thanks for your quick response**
> > >
> > > We appreciate you so much for the quick response, and we are delighted to see your approval of our revision to this paper.
> > >
> > > Prompted by your suggestion, we added Appendix C to compare the parameter complexity of our method with other methods. In this paper, as the second major contribution, we propose detaching feature transformation from graph convolution,  reducing the parameter complexity remarkably, e.g., from $O(n*p*q)$ to $O(n+p*q)$ for Spectral CNN and our GWNN. Such a practice offers us an efficient implementation of GWNN, making it applicable to large scale real world networks. Moreover, with the reduction of parameter complexity, GWNN is particularly appropriate for the scenario of semi-supervised learning where labeled data is limited. For example, for the graph-based supervised learning task, i.e., node classification on Cora and Citeseer, GWNN with smaller parameter complexity achieves better performance than ChebyNet.
> > >
> > > Thank you very much for all your comments, which are valuable for improving and strengthening our paper.

---

> > > > ### Comment · AnonReviewer1 · 2018-11-30
> > > > **-**
> > > >
> > > > Thank you for your answer. I updated my score as promised.
> > > >
> > > > Regards,

---

> > > > > ### Author Response · Authors · 2018-12-03
> > > > > **Thank you for your response**
> > > > >
> > > > > Thank you very much for your efforts on reviewing the paper and the response. Your review comments are valuable for improving and strengthening our paper. We are delightful to see that our revisions are acknowledged by you.
> > > > >
> > > > > Regards,
> > > > > The authors

---

> ### Author Response · Authors · 2018-11-19
> **Response to Reviewer1 (1/2)**
>
> Thank you for your positive comments and the accurate summary of our contributions. Prompted by your constructive suggestions, we carefully revised the paper and improved the clarity of the presentation. Next we offer point-by-point response.
>
> Q1: (1) « Some subsequent works devote to making spectral methods spectrum-free… avoiding high computational cost ». I think also those types of representations can be potentially unstable. That’s a second reason.
>
> A1: We revised this sentence in the new version, highlighting that the chief goal of spectrum-free methods is to achieve locality in spatial domain.
>
> Q2: I’m not sure to understand the point (1) of the fourth paragraph of the introduction. (1.)« graph wavelet does not depend on the eigen decomposition of Laplacian matrix ». Does it mean numerically ? This sentence is not clear, because even numerically, it can be done in a dependent way to this matrix. However, if it is implied that the fastest algorithm can be obtained without eigen-decomposition of the Laplacian Matrix, in a cheap way, then I agree and a small rephrasing could be nice.
>
> A2: We apologize for not clarifying this point at the outset. To be sure, as you pointed out, graph wavelets are “numerically”, not intrinsically, independent on eigen-decomposition of Lapacian matrix. For clarity, we replaced the sentence “graph wavelet does not depend on the eigen-decompostition of Laplacian matrix” with a new sentence “graph wavelets can be obtained via a fast algorithm without requiring the eigen-decomposition of Laplacian matrix”.
>
> Q3: Please remove all the sentences that are supposed to be an introduction for a section, i.e. the sentences between 2 and 2.1 (« We use graph… ») and 3 and 3.1 (« In many research… »). They are poorly written and do not help the reader.
>
> A3: Thank you for the suggestions. We removed these sentences in the revised version.
>
> Q5: .(2.2) “However, such a polynomial approximation also limits the flexibility to define appropriate convolution on graph” I’m not sure to understand. In the paper you refer to, the set of filters span the polynomial of degree less than n, of a diagonalizable matrix of size nxn. Thus, lagrange polynomial basis could be used to interpolate any desired values? Does it signify learning non-diagonal(in the appropriate basis) matrix?
>
> A5: We apologize for the misleading statement. Indeed, a polynomial parameterization with order $n$ is capable to represent any diagonal matrix, i.e., graph convolution kernel in spectral domain. What we mean by “the inflexibility of such polynomial parameterization” is: a smaller order causes approximation bias and a larger order results in non-localized convolution kernel. In the revised version, we clarified this point in the last paragraph in Section 2.2.
>
> Q6: (2.3) $s$ how is this parameter chosen, crossvalidation? Is it adjusted such that the singular values of $\psi_s$ have a certain decay? Why is $s$ constant across layers? How is the spectrum of psi_s? Is it well conditionned? A huge difference with (Hammond et al,2011) is that they use a collection of wavelets. Did you consider this kind of approach ? Is there a loss of information in some cases?(like if $\psi_s$ has a fast decay)
>
> A6: The parameter $s$ is a hyper-parameter in our method, and its value is chosen using cross-validation. We use a constant $s$ across layers in this paper. We agree with you that it is a really interesting idea to use different $s$ across layers, i.e., using wavelets with varying locality across layers. Different form previous methods that use a collection of wavelets, we use the wavelets associated with a constant scaling parameter $s$. We agree that it is promising to use a collection of wavelets and leave it as future work. In our paper, the larger the $s$ is, the faster decay the $\psi_s$ has, moreover, the range of neighboring node is large, which may result in the loss of localization. Finally, in the revised version, we included detailed discussion in Appendix B to demonstrate the influence of the value of $s$ on the performance of our method.

---

### Official Review · AnonReviewer3 · 2018-11-05
**Long-awaited combination of Graph wavelets and NNs**

**Rating:** 7
**Confidence:** 5

**Review:**

This paper proposes to learn graph wavelet kernels through a neural network. This idea is interesting, even if it is not a real surprise, given the interesting features graph wavelets, and the explosion of proposals for graph neural networks. Yet, the paper is interesting, pretty complete, and the proposed algorithm is shown to be relatively effective.

A few more detailed comments:

- one for the key motivations the work, namely to avoid eigen-decompositions, is also solved in a different way by methods using Chebyshev approximations (e.g., Khasanova - ICML 2017). The introduction should be clarified accordingly.
- retaining the flexibility of convolution kernel (on page 2): what do the authors mean here?
- the graph wavelet nn in 2.4 is certainly an interesting combination of known elements - yet, it is not very clear how the network is trained/optimized at this stage.
- the section 3 is a bit confusing: are the proposed elements, contributions of the paper, or only ideas with future work?
- the idea of detaching feature transformation from convolution is interesting: it would be even better to quantify or discuss the penalty, if any, induced by this design choice.
- the results are generally fine and convincing, even if they are not super-impressive. ' GWNN is comfortably ahead of Spectral CNN. ' is probably an over-statement however...
- the discussion about interpretability is interesting, and very trendy. However, what the authors discuss is mere localisation and sparsity - this is one way to 'interpret' interpretability of features, but that discussion should be rephrased in a more mild sense then.

Generally, the ideas in this paper are interesting, even not surprising. The text should be clarified at places, and the paper looks a bit superficial on many aspects (see above). With good revision, it could lead to an interesting paper, and most likely to interesting discussions at ICLR.

---

> ### Author Response · Authors · 2018-11-19
> **Response to Reviewer3**
>
> Thank you for the pertinent comments and accurately summarizing the major contributions of this paper. According to your constructive suggestions, we carefully revised our paper with much improvement. We now offer point-by-point response.
>
> Q1: one for the key motivations the work, namely to avoid eigen-decompositions, is also solved in a different way by methods using Chebyshev approximations (e.g., Khasanova - ICML 2017). The introduction should be clarified accordingly.
>
> A1: We revised the statements relevant to this point, and cited the paper (i.e., Khasanova, ICML 2017) in the introduction part.
>
> Q2: retaining the flexibility of convolution kernel (on page 2): what do the authors mean here?
>
> A2: We apologize for the misleading statement. What we mean is that the convolution kernel of ChebyNet is not flexible. Specifically, ChebyNet offers a $K$-order polynomial parameterization to graph convolution kernel. A smaller $K$ causes high approximation bias, while a larger $K$ results in non-localized convolution kernel. Therefore, ChebyNet has limited flexibility to define graph convolution kernel. In the revised version, we clarified this point in the last paragraph in Section 2.2.
>
> Q3: the graph wavelet nn in 2.4 is certainly an interesting combination of known elements - yet, it is not very clear how the network is trained/optimized at this stage.
>
> A3: Thank you for pointing out this issue. Prompted by your suggestion, we revised Section 2.4: (1) We offered detailed description about the architecture of graph wavelet neural network; (2) We added the loss function when training graph wavelet neural networks on semi-supervised node classification task.
>
> Q4: the section 3 is a bit confusing: are the proposed elements, contributions of the paper, or only ideas with future work?
>
> A4: We apologize for the confusing organization of sections. Indeed, Section 3 is the second contribution of this paper, i.e., reducing the parameter complexity by detaching feature transformation from graph convolution. It is particularly important for training graph wavelet neural networks. For clarity, we reorganized the sections of this paper: combing the original Section 3 into Section 2 as a subsection, i.e., Section 2.5.
>
> Q5: the idea of detaching feature transformation from convolution is interesting: it would be even better to quantify or discuss the penalty, if any, induced by this design choice.
>
> A5: We are delighted to see that this idea is identified and approved. Detaching feature transformation from convolution, we remarkably reduce the number of parameters. This practice is particularly important for scenarios where labeled data is limited, e.g., graph-based semi-supervised learning task considered in this paper.
>
> One potential penalty of this practice is that the modeling capacity is reduced. To quantify the penalty, in Section 4.4, we compared the influence of detaching feature transformation from convolution to the performance of graph-based semi-supervised node classification. Results demonstrate that detaching feature transformation from convolution achieves comparable (sometimes better) performance, with the number of parameters remarkably reduced.
>
> Q6: the results are generally fine and convincing, even if they are not super-impressive. ' GWNN is comfortably ahead of Spectral CNN. ' is probably an over-statement.
>
> A6: For experimental evaluation, we compare the proposed GWNN with existing spectral methods, Spectral CNN, ChebyNet, GCN and some spatial methods like MoNet. The major contribution of this paper is to improve spectral methods, using graph wavelets rather than eigenvectors of Laplacian as bases. Thus, we focus on the comparison with spectral methods. By the sentence ' GWNN is comfortably ahead of Spectral CNN ', we mean GWNN using graph wavelet transform is ahead of the Spectral CNN using Fourier transform, i.e., GWNN (the last row in Table 3) and Spectral CNN (the ninth row in Table 3). Indeed, GWNN is better than Spectral CNN by 10% improvement on Cora and Citeseer and 5% improvement on Pubmed.
>
> Q7: the discussion about interpretability is interesting, and very trendy. However, what the authors discuss is mere localisation and sparsity - this is one way to 'interpret' interpretability of features, but that discussion should be rephrased in a more mild sense then.
>
> A7: Thank you for the inspiring comments. We revised the discussion about interpretability (Section 4.7), trying to ease the understanding of the interpretability of graph wavelets and graph convolution via graph wavelet transform.

---

### Official Review · AnonReviewer4 · 2018-11-07
**Promising but need further development**

**Rating:** 4
**Confidence:** 4

**Review:**

This paper proposed a new formulation of graph convolution, that is based on
graph wavelet transform. The convolution network is expressed in eq.(4)
with \psi_s given by eq.(1). This new formulation is exciting in that
it has numerous advantages comparing to spectral graph convolutions
such as the sparsity of \psi_s (see the items listed in section 2.3).
The method showed marginal improvement (<1%) on node classification
tasks of citation networks.

Overall, I am convinced with the technical novelty and that the method
can be promising. However, in its current form, there are several
major weaknesses, hinting that this work needs further developments before
publication.

1. By the vanilla implementation without any approximations, the multiplication
with \phi_s and \phi_s^{-1} has quadratic complexity, and to compute these two
matrices is of cubic complexity.
This is not acceptable for large graphs. In section 3.2, the authors
mentioned that Hammond et al (2011) have an approximation of
\phi_s and \phi_s^{-1} with the complexity of O(|E|).
However, this important technical detail is disappointingly missing
in the main text. I have further concerns about whether the good
properties listed in section 2.3 are preserved by this approximation,
which is not discussed in the paper.

Even with such an approximation, the matrix multiplication still
has quadratic complexity. To reduce this complexity needs some non-trivial
developments. I suggest the author(s) dive into the expression of
\phi_s and write the convolution in the vertex domain and seek possibilities
for a linear approximation.

2. Experimentally, the improvement over GCN is marginal. Taking into account
the implementation difficulty and complexity, the
proposed method is more like a proof-of-concept instead of
being of practical use. The authors are suggested to make the
empirical study more comprehensive, by including node classification
in an inductive setting, and/or including link prediction experiments.

3. The hyper-parameters (scale of the heat kernel) and t (threshold
to zero the \phi_s matrix) has to be tuned for each data set.
This is not the ideal case because these parameters may not be easy
to tune for real data sets, making the method difficult to use.
The authors should at least give some recipes on how to tune these parameters.

---

> ### Author Response · Authors · 2018-11-19
> **Response to Reviewer4 (2/2)**
>
> Q2: Experimentally, the improvement over GCN is marginal. Taking into account the implementation difficulty and complexity, the proposed method is more like a proof-of-concept instead of being of practical use. The authors are suggested to make the empirical study more comprehensive, by including node classification in an inductive setting, and/or including link prediction experiments.
>
> A2: In this paper, we focus on spectral methods for graph convolution. The standard spectral method, i.e., the Spectral CNN, is not localized, limiting its performance. To achieve localization, GCN and ChebyNet are proposed to express the graph convolution defined via graph Fourier transform in vertex domain. Here, we propose a new formulation of graph convolution, i.e., defining graph convolution via graph wavelet transform, achieving localization in both spectral and spatial domain.
>
> Experimental results demonstrate that the proposed GWNN method using graph wavelet transform remarkably (achieving 10% improvement on Cora and Citeseer, and 5% improvement on Pubmed [Table 3]) outperforms the Spectral CNN method using graph Fourier transform. Meanwhile, GWNN also outperforms GCN and ChebyNet.
>
> We fully agree that it is always better to validate a new method on more scenarios and tasks. Here, following the common practice to evaluate spectral methods (e.g., GCN and ChebyNet) for graph CNN, we validate our method on the widely-used playground, i.e., node classification task on three benchmark datasets.
>
> Q3: The hyper-parameters (scale of the heat kernel) and t (threshold to zero the \phi_s matrix) has to be tuned for each data set. This is not the ideal case because these parameters may not be easy to tune for real data sets, making the method difficult to use. The authors should at least give some recipes on how to tune these parameters.
>
> A3: The hyper-parameter $s$ is used to modulate the range of neighborhood and the smoothness of graph wavelets. The hyper-parameter $t$ is used only for computational consideration. We use cross-validation to determine the value of the hyper-parameter $s$ and $t$, following the common practice of machine learning community. To offer more intuition, we add the analysis about the impact of hyper-parameters on the accuracy of graph-based semi-supervised learning in Appendix B, and demonstrate our recipes to tune hyper-parameters.

---

> > ### Comment · AnonReviewer4 · 2018-11-26
> > **Reply to rebuttal**
> >
> > Thank you very much for the detailed clarification and revision on parameter sensitivity (the added Figure 5 is very informative).
> >
> > I will keep my original scores based on
> >
> > - Hammond et al (2011)'s O(|E|) approximation of \phi_s and \phi_s^{-1} are still missing.
> >
> > - It is true that the proposed GWNN has some computational issues and is surely more expensive than GCN (that's why you need Hammond et al's approximation) and has more hyperparameters resulting from Hammond's polynomial approximation as well as the heat kernel. I am not convinced that the authors have fully acknowledge this point in their response.
> >
> > - Experimentally, WAVELET has around 1% improvement over GCN's originally reported results, which is a fair baseline. Actually, GCN's performance can be a bit better than the reported results. For example, on the Pubmed dataset, its transductive classification accuracy can reach 79.2%, which is better than the proposed method. Moreover, as other reviewers/researchers have pointed out, the inductive results are missing.
> >
> > - In the sparsity comparison, one should also compare with the sparsity of the graph Laplacian matrix, so as to have an idea on the computational overhead of using the proposed approach.

---

> > > ### Author Response · Authors · 2018-11-27
> > > **Response to Reviewer4 (2/2)**
> > >
> > > Q3: Experimentally, WAVELET has around 1% improvement over GCN's originally reported results, which is a fair baseline. Actually, GCN's performance can be a bit better than the reported results. For example, on the Pubmed dataset, its transductive classification accuracy can reach 79.2%, which is better than the proposed method. Moreover, as other reviewers/researchers have pointed out, the inductive results are missing.
> > >
> > > A3: Thank you for letting us to know that GCN can achieve better result (i.e., 79.2%) on Pubmed with well-tuned parameters. Indeed, GCN is also not the state-of-the-art methods for transductive node classification task. Several spatial methods like GAT achieve better results than GCN. We didn’t compare with GAT in this paper, because we focus on the line of spectral methods.
> > >
> > > In other words, we don’t aim to design state-of-the-art method for graph-based semi-supervised learning task, e.g., node classification. We aim to demonstrate that graph wavelet transform is better than graph Fourier transform for designing spectral graph convolution. As shown in Table 3, the proposed GWNN method using graph wavelet transform remarkably (achieving 10% improvement on Cora and Citeseer, and 5% improvement on Pubmed) outperforms the Spectral CNN method using graph Fourier transform.
> > >
> > > For the comparison with GCN, we want to clarify the connection and difference between our method and GCN. Both our method and GCN aim to improve spectral methods via design localized graph convolution. GCN, as a simplified version of ChebyNet, expresses the spectral graph convolution in spatial domain, acting as spatial-like method (Monti et al., 2017). Our method resorts to using graph wavelet as a new set of bases, directly designing localized spectral graph convolution. We compare with GCN on the transductive node classification task, just showing that these two methods have comparable results.
> > >
> > > Finally, spectral methods are inappropriate for the inductive classification task. Indeed, before GraphSAGE and GAT, almost all methods are not evaluated on inductive task. We follow the practice of spectral methods and evaluate our methods on the transductive classification task.
> > >
> > > Q4: In the sparsity comparison, one should also compare with the sparsity of the graph Laplacian matrix, so as to have an idea on the computational overhead of using the proposed approach.
> > >
> > > A4: We fully understand your concern about the computational cost of the proposed method. We acknowledge that the sparsity of the graph wavelet matrix depends on the sparsity of the Laplacian matrix and the hyper-parameter $s$. For clarity, we added the sparsity comparison between the two matrices in Appendix E.
> > >
> > > For the computational overhead, it is not necessary to explicitly obtain the graph wavelet matrix for graph wavelet transform. Instead, following the Chebyshev approximation in Eq. (17) in Appendix D, the computational complexity of graph wavelet transform is O(m*|E|).

---

> > > > ### Comment · AnonReviewer4 · 2018-12-03
> > > > **Further reply**
> > > >
> > > > I would like to thank the authors for the detailed reply and the revision efforts. After reading the rebuttals, I think the authors have established a mutual understanding with me of the strength and weakness of the paper. Our difference is more on "whether the strength is great enough" and "whether these weaknesses are OK" for giving the paper a pass.
> > > >
> > > > After careful consideration and re-scan through the paper, I still would like to keep my score. I understand that the authors have made revisions that should be acknowledged as these revisions have improved the presentation.  However, these revisions and discussions have also confirmed the weakness as pointed out in my original review.
> > > >
> > > > Let me summarize my review as follows
> > > > Pro
> > > > - technical novelty
> > > > - presentation quality
> > > > Con
> > > > - computational and implementation difficulty
> > > > - marginal improvement over GCN
> > > >
> > > > I would encourage the authors to develop this method into the next stage. Based on the novelty, I believe it deserves to be published in a good venue as ICLR after some further developments.

---

> > > > > ### Author Response · Authors · 2018-12-03
> > > > > **Reply to Reviewer4**
> > > > >
> > > > > We appreciate you so much for the pertinent comments. With the two-round rebuttals and revisions, we are inspired that we have established a mutual understanding with you about the paper.
> > > > >
> > > > > Considering that you still have some concerns, we would like to highlight our major improvements after revisions:
> > > > >
> > > > > - Informative analysis about hyper-parameter sensitivity.
> > > > > Figure 5 is added as a thorough analysis about the effect of the hyper-parameter on the performance of the proposed method. As shown in Figure 5, the proposed method is not sensitive to hyper-parameters.
> > > > >
> > > > > - Clarification about computational complexity.
> > > > > Prompted by your suggestions, we clarified the computational complexity of the graph wavelets and graph convolution via graph wavelets. Indeed, with Hammond’s approximation (Appendix D), the computational complexity scales up to O(|E|) rather than being of quadratic complexity as concerned in your original review.
> > > > >
> > > > > - Good presentation quality.
> > > > > We carefully revised the paper, clarifying several misleading statements in the original version. Moreover, we added detailed description about the implementation of the proposed method (Sec. 2.4).
> > > > >
> > > > > We also agree that our method has several limitations, e.g., higher computational cost than GCN. However, these limitations don’t deteriorate our major contribution, i.e., we offer a localized graph convolution via replacing graph Fourier transform with graph wavelet transform, and such a work is useful, at least promising, to help us design appropriate graph convolution via finding good spectral basis with localization property and good interpretability.
> > > > >
> > > > > In sum, we are delighted to see that the technical novelty and presentation quality are acknowledged. With the aforementioned improvements, we believe the current version of this paper deserves a higher score than the original version.

---

> > > ### Author Response · Authors · 2018-11-27
> > > **Response to Reviewer4 (1/2)**
> > >
> > > Thank you for your efforts on reviewing our paper and the response. We are sorry that our hard-working response fails to meet your requirement. Here, we offer more response and revise the paper with some supplements.
> > >
> > > Q1: Hammond et al (2011)'s O(|E|) approximation of \phi_s and \phi_s^{-1} are still missing.
> > >
> > > A1: We acknowledge that it would be much self-contained for the paper to include the technical details of the approximation of \phi_s and \phi_s^{-1}. Considering that the approximation algorithm isn’t proposed by this paper, in the previous version, we only give the main results (e.g., complexity analysis) of the approximation with the reference to Hammond et al (2011). Prompted by your request, we included the technical details of the approximation of \phi_s and \phi_s^{-1} as Appendix D in the new version.
> > >
> > > Q2: It is true that the proposed GWNN has some computational issues and is surely more expensive than GCN (that's why you need Hammond et al's approximation) and has more hyper-parameters resulting from Hammond's polynomial approximation as well as the heat kernel. I am not convinced that the authors have fully acknowledged this point in their response.
> > >
> > > A2: Thank you for the pertinent comments. We also acknowledge that the proposed method still has some limitations, e.g., the computational issues and additional hyper-parameters as you pointed out, although we have attempted to combat these issues via approximation and detaching feature transformation from graph convolution. Yet, the major contribution, i.e., offering a localized graph convolution via replacing graph Fourier transform with graph wavelet transform, is not deteriorated by these issues. In particular, such a work is useful, at least promising, to help us design appropriate graph convolution via finding good spectral basis with localization property and good interpretability. This distinguishes our method from ChebyNet and GCN, which express the graph convolution defined via graph Fourier transform in vertex domain. In the revised version, we acknowledged these issues and added Appendix B and E to discuss the hyper-parameter and the computational cost.

---

> ### Author Response · Authors · 2018-11-19
> **Response to Reviewer4 (1/2)**
>
> Thank you for the positive comments. We are delighted to see that the technical novelty of the proposed method is approved. Inspired by the pertinent comments, we carefully revised the paper, clarifying some misleading statements and highlighting the main contributions.
>
> Q1: By the vanilla implementation without any approximations, the multiplication with \phi_s and \phi_s^{-1} has quadratic complexity, and to compute these two matrices is of cubic complexity. This is not acceptable for large graphs. In section 3.2, the authors mentioned that Hammond et al (2011) have an approximation of \phi_s and \phi_s^{-1} with the complexity of O(|E|). However, this important technical detail is disappointingly missing in the main text. I have further concerns about whether the good properties listed in section 2.3 are preserved by this approximation, which is not discussed in the paper.
>
> Even with such an approximation, the matrix multiplication still has quadratic complexity. To reduce this complexity needs some non-trivial developments. I suggest the author(s) dive into the expression of \phi_s and write the convolution in the vertex domain and seek possibilities for a linear approximation.
>
> A1: We apologize for not clarifying these issues at the outset. We appreciate you for pointing out these issues. Prompted by your comments, we carefully revised Sec 2.3 to ease the understanding of the property of graph wavelets. Here, we clarify three main points:
>
> (1) The complexity of the multiplication with \phi_s and \phi_s^{-1} is not quadratic. Indeed, the complexity depends on the sparsity of the two matrices \phi_s and \phi_s^{-1}. For real world networks, the number of edges is much less than the square of the number of nodes. As a result, as proved in Lemma 5.5 in Hammond et al (2011), both \phi_s and \phi_s^{-1} are sparse. Table 4 also shows the empirical results about the sparsity of \phi_s^{-1}: on the dataset Cora, more than 97% elements in \phi_s^{-1} are zero.
>
> (2) The computation of the two matrices is completed using a polynomial approximation, Sec. 6 in Hammond et al (2011). Such an approximation is a polynomial function of Laplacian matrix, only involving the matrix-vector multiplication. Since the sparsity of the Laplacian matrix scales up to O(|E|), the computation of \phi_s and \phi_s^{-1} is of the complexity O(K*|E|), where |E| is the number of edges and K is the order of polynomial approximation. This point is clarified in Sec. 2.3.
>
> (3) All the properties listed in Section 2.3 are satisfied when \phi_s and \phi_s^{-1} are computed in an approximated way: (a) The first property (i.e., high efficiency) is naturally satisfied; (b) The second property (i.e., high sparseness of \phi_s and \phi_s^{-1}) is satisfied, since the approximated \phi_s and \phi_s^{-1} are always more sparse than the original ones; (c) The third property (i.e., localization property) is satisfied. As shown in Appendix A, the localization property is a result of the sparseness of graph wavelets; (d) The forth property (i.e., flexible neighborhood) is satisfied. Indeed, the flexibility is achieved by adjusting the value of the scaling parameter $s$.
>
> In addition, we agree it is a really interesting idea to write the convolution in the vertex domain and seek possibilities for a linear approximation. Indeed, we are attempting to use heat kernel to achieve this purpose. Here, we leave it as future work.

---

### Public Comment · (anonymous) · 2018-09-30
**Not report the results of graph attention network (GAT)**

You should include the experimental results of GAT which are better than yours and then explain why your results are lower.

---

> ### Author Response · Authors · 2018-10-01
> **Thank you for your feedback**
>
> Thank you for your comment! I would like to explain it from two perspectives:
> Motivation: Due to the inspiration of Spectral CNN (Bruna et.al., 2014), many works attempt to implement convolutional neural networks on graphs. However, as far as I know, there are few spectral methods leveraging the convolution theorem to design convolution operator after GCN (Kipf&Welling, 2016). It's true that GAT is more flexible than spectral methods, it applies attention mechanism to calculate relations between nodes. However, the shared parameters in GAT are linear transformation matrix W when calculating attention instead of convolution kernels (filters). We hope to propose a better spectral method based on the convolutional theorem. GWNN is our first attempt which holds the flexibility of kernel and locality in vertex domain.
> Results: We think we achieve comparable results on two datasets. The accuracy of GAT on Cora is 83.0%, and that of our work is 82.8%. On Pubmed, GAT scores 79.0% while our work scores 79.1%. It's true that GAT performs better than ours on Citeseer. We think it could be caused by the fact that the network of Citeseer is sparser than the other two datasets, which has 3327 nodes and 4732 edges. And due to the way of GAT which calculates the relations between nodes based on representation in hidden layer, adjacency matrix is only regarded as a mask. Instead, we leverage the wavelet which depends on the structure of network. Thus, our model doesn't achieve a better results on Citeseer.
> Thanks again for your comment! We will include GAT as our baseline and give a discussion in a revised version of our paper.

---

> > ### Public Comment · ~Petar_Veličković1 · 2018-10-01
> > **Supervised state-of-the-art**
> >
> > Disclosure: I'm the author of GATs, and I'm not the original comment poster from above.
> >
> > Because the purpose of this paper seems to be to propose a fully-spectral method, I agree that the main comparison should be with spectral methods, or at least methods that depend on observing some aspects of the Laplace matrix (i.e. Chebyshev networks, GCNs and the original formulation of MoNet).
> >
> > To the best of my knowledge, the state-of-the-art (at least on Citeseer, which seems to be the most flexible dataset to improve on out of the three) is currently held by GraphSGAN (Ding et al., CIKM 2018), and it might be useful to report this result instead (but this would only be for contextual purposes, in my opinion).

---

> > > ### Public Comment · ~Marc_Brockschmidt1 · 2018-10-01
> > > **Value of experiments on CORA/Citeseer/Pubmed datasets**
> > >
> > > Disclosure: I'm one of the authors on the GGNN/GPNN papers, and I don't want to comment on the merits of the presented approach here, but on problems with graph learning experiments overall.
> > >
> > > Having done some experiments with GNN variants on the Cora/Citeseer/Pubmed, I fear they are not useful indicators of the usefulness of new ideas anymore. The results reported by recent work are all very near to each other, and differences seem to be mostly noise. For example, GPNN reports 79.3% acc on Pubmed and 81.8% on Cora, but only 69.7% on Citeseer. These GPNN results are averaged over a number of runs with different seeds, but the variance is substantial compared to the differences between published models. Concretely, I believe that a few rounds of optimizing the 'SEED' hyperparameter would be most likely sufficient to get any of the models published in the last year to have the best results.
> > >
> > > What I'm saying here has nothing to do with the merits of any of the papers on graphs submitted to ICLR'19, and I'm not calling for error bars on the results here. Instead, I think we had enough progress that we /all/ need to move on to harder benchmarks, though I'm not sure what these should be. The chemical properties from the Gilmer et al. paper may be a good fit (and the data is easily accessible and reference implementations exist), but they are all very small graphs. Our graph data from the Allamanis et al. ICLR'18 paper has been released and contains larger graphs, but the dataset is painfully large to work with and there's many non-graph things to play with.

---

> > > > ### Author Response · Authors · 2018-10-01
> > > > **Our opinions about datasets and methods**
> > > >
> > > > Thank you for your comment! It’s interesting to have a discussion on the datasets.
> > > >
> > > > Up to now, GNN has got a lot of attention, and many researchers are making contributions to this hotspot. We agree that harder benchmarks could benefit the development of GNN. However, it’s still a long way to give an appropriate definition of the relation between nodes no matter what the benchmarks. Before the emergence of a more convincing dataset, we have to validate the proposed models on these widely adopted ones. And experiments on the three datasets help us to analyze the relative merits of different models to some degree.
> > > >
> > > > In our opinion, conducting experiments is just a way to validate the merits and effectiveness of a new proposed model. And it's more valuable to have a theoretical exploration in addition to achieving better performance on the datasets. These related works mentioned in the paper provide different ideas and promote the development of GNN. As mentioned in the paper, one of our main contributions is to point out the value of wavelet and localized basis.
> > > >
> > > > I agree that we need to pay attention to harder benchmarks. Our group also focuses on social network analysis. And we are trying to generalize our method to social networks, and validate the effectiveness of different models with related benchmarks. Also, we will pay attention to the datasets and works that you mentioned.
> > > >
> > > > Thanks again for your comment!

---

> > > ### Author Response · Authors · 2018-10-01
> > > **Thanks for your comment**
> > >
> > > Thanks for your comment and approval of our baseline!
> > > GAT has become a popular work, and there are many variants of GAT. Although our work is a spectral method, we still get inspiration from the attention mechanism. Also, we will pay attention to GraphSGAN.
> > > Thank you very much!

---

### Public Comment · ~Michael_Bronstein1 · 2018-10-01
**baselines and benchmarks**

I think this is an interesting work, and I would like to follow up on the previous exchange of comments.

First, I would like to note that the distinction between "spectral" and "spatial" approaches is rather artificial, as at the end methods like Chebyshev networks or the present paper do not do explicit Fourier transform and boil down to applying local spatial operators (e.g. Laplacian and its powers).

However, if by "spectral" methods one refer to those based on Laplacian-type operators, I would suggest comparing to the following baselines: [1] uses rational functions instead of polynomials (it achieves 81.9% on the standard Cora split). One of the key deficiencies of the Laplacian is that it is locally permutation-invariant (on a plane, this is manifested as rotation symmetry). Thus, any Laplacian-based filters are isotropic. It is possible to create anisotropic spectral filters on manifolds, but general graphs are more complicated. I am only aware of [2] that uses graph motifs (motivated by the work of Benson et al.) to create an analogy of anisotropic diffusion on graphs. This is especially useful for treating directed graphs (as a matter of fact, the original Cora citation graph is directed), which are somewhat problematic to treat with spectral techniques due to the asymmetry of the Laplacian matrix.

Among "spatial" methods, besides GAT you might want to check [3], which alternates convolutions on vertices and edges (using the formalism of line graphs), generalizing the graph attention mechanism proposed in GAT and leading to better performance, though not very significantly (achieving 83.3% on Cora and 72.6% on Pubmed). Also, [4] is an interesting approach based on graph shift operators (in general, the works of Jose Moura on graph signal processing are unjustly not cited in this community).


Second, in the context of shape analysis we developed a local spectral CNN approach based on the graph Windowed Fourier Transform [5], which bears some resemblance to your paper (though never tested it on general graphs).


Third, I would add my 50 cents to Marc's comment regarding benchmarks: I also think Cora and Pubmed are "too easy". Even worse, they might provide a misleading idea about how different methods perform on "real data". From my experience, most of GCNN approaches work well on graphs with underlying assumption of homophilic relations ("positive connections"). This in particular true for geometric data sampled from some high-dimensional manifolds. Laplacian-based methods are very appropriate in these settings. It seems that Cora/Pubmed citation networks fall into this category. However, more challenging datasets (such as interactomes in system biology) might have more complicated heterogenous relations between nodes on which algorithms working well on Cora perform poorly. What is also blatantly missing are interesting datasets with rich edge features. It seems to be sufficient critical mass of works on graph deep learning to motivate the creation of more challenging benchmarks.


1. CayleyNets: Graph convolutional neural networks with complex rational spectral filters", arXiv:1705.07664

2. MotifNet: a motif-based Graph Convolutional Network for directed graphs", arXiv:1802.01572

3. Dual-Primal Graph Convolutional Networks, arXiv:1806.00770.

4. ON GRAPH CONVOLUTION FOR GRAPH CNNS, DSW 2018

5. Learning class-specific descriptors for deformable shapes using localized spectral convolutional networks", Computer Graphics Forum 34(5):13-23, 2015

6. Learning shape correspondence with anisotropic convolutional neural networks, NIPS 2016.

---

> ### Author Response · Authors · 2018-10-02
> **Response to baselines and benchmarks**
>
> Thank you for the comprehensive and pertinent comments. We also appreciate you for listing several recent advances, and we will include these works in related works or as baseline methods.
>
> First, for the difference between “spectral” and “spatial” methods, we basically agree with you that their distinction is somewhat artificial. To our understanding, the major distinction lies in the way the convolution is defined rather than whether Fourier transform or wavelet transform, i.e., transforming graph signal from vertex domain to spectral domain, is explicitly leveraged. Indeed, many spectral methods are spectrum-free, e.g., Chebyshev networks and GCN. In this paper, we separately describe the two types of methods just to put our graph wavelet neural network in an appropriate literature, i.e., moving in line with spectral methods.
>
> We also fully agree that it is a promising research direction to design anisotropic spectral filters on graphs, following the successful practice on manifolds.
>
> Second, we are glad to see that you have a local spectral CNN approach based on the graph Windowed Fourier Transform. We believe that this is an interesting work, given that Windowed Fourier transform is localized in vertex domain, compared with Fourier transform. This is also why we want to use wavelet transform to replace Fourier transform.
>
> Third, a new benchmark dataset is really an interesting topic and urgent demand for the GNN research community, through our present work is still evaluated on the three traditional benchmark datasets, i.e., Cora, Citeseer and Pubmed. To our understanding, the current benchmark works well to distinguish good methods/models from bad ones, yet they are not sufficient to distinguish better methods/models from good ones. We also look forward to seeing the emergence of new benchmark. The other issue we want to spell out is that a new benchmark dataset is highly dependent on new task or scenario, e.g., the current benchmark is basically designed for graph semi-supervised classification task and the underlying assumption is the graph smoothness, i.e., connected nodes are likely to share the same label. We believe that a new task or a more heterogeneous scenario will be valuable to promote the development of this research community.
>
> In sum, thank you for the comments and inspiring discussion again.

---

### Author Response · Authors · 2018-11-19
**Summary of revisions made to the paper**

We thank the reviewers and other researchers for their comprehensive and thoughtful comments. According to these comments, we carefully revised our paper, improving both the quality and clarity of our paper. We hope that reviewers will find the revised paper suitable for acceptance.

A summary of the changes made to our paper is listed as follows:

-	In Section 2.2, we added a detailed discussion about why ChebyNet has limited flexibility to define convolution kernel. This revision clarifies the confusion about the “the flexibility of convolution kernel” (Reviewer 1 and Reviewer 3).

-	In Section 2.4, we added the architecture of the proposed graph wavelet neural network and the loss function when training graph wavelet neural networks on semi-supervised node classification task.

-	We combined Section 3 in the original version into Section 2.5 in the revised version (Reviewer 3).

-	We added an experiment (Appendix B) to show the impact of hyper-parameters on the accuracy of node classification, and demonstrate our recipes to tune hyper-parameters (Reviewer 1 and Reviewer 4).

-	We rephrased some misleading statements to improve the clarity of the paper.

-	We added some references, prompted by the suggestions from reviewers and public comments.

We submitted a revised version including the aforementioned revisions. Thank you for all the efforts that help us improve the paper.

---

### Public Comment · (anonymous) · 2018-12-01
**Interesting study using graph wavelets**

This is an interesting study. In particular, training wavelets on graphs is very useful. It is interesting to compare with https://arxiv.org/abs/1804.00099. The latter work also uses Hammond's wavelets, but instead of training the wavelets it uses them to construct a scattering transform. It can be regarded as a graph network with fixed parameters. It is proved to have properties such as invariance to permutation and stability to signal and graph manipulation.

---

> ### Author Response · Authors · 2018-12-05
> **Thanks for your comment**
>
> Thank you for your comment! This paper seems interesting, i.e., any feature generated by the network is
> approximately invariant to permutations and stable to graph manipulations.  We will pay attention to it and add it as our related work if necessary.

---

### Public Comment · ~Benedek_Rozemberczki1 · 2019-01-24
**Attempt to reproduce results.**

I attempted to reproduce the results. The accuracy scores are lower than reported, even with a larger training set. What was your exact approach to defining/creating a wavelet filter?

https://github.com/benedekrozemberczki/GraphWaveletNeuralNetwork

---

> ### Author Response · Authors · 2019-01-24
> **Thank you for your comment**
>
> Thank you for your feedback and reproduce!
>
> I implemented my code via tensorflow. In my code, I set the elements which are lower than a given threshold to zero, which is stated in the paper. Also, the datasets are divided as GCN (https://github.com/tkipf/gcn). I will organize and release my code in Github as soon as possible, and update the link of Github here.

---

> > ### Public Comment · ~Benedek_Rozemberczki1 · 2019-01-30
> > **Thank you for the code release.**
> >
> > I saw the code release. Linked it in the readme of the repository.

---

### Public Comment · (anonymous) · 2019-10-16
**Questions about the definition of the graph wavelet transform in this paper!**

Compared with GCN,  this paper replaced the non-localized and non-sparse graph Fourier bases with the so-called "graph wavelet bases" ,  leveraging the sparseness and locality of graph wavelet bases to improve the efficiency of GCN. However, I have some questions about the definition of the graph wavelet transform in this paper.

First, in section 2.3, you defined $\phi_s$ as the graph wavelet bases, and the kernel is $g(s\lambda)=e^{s\lambda}$ . However,  it is certainly not the graph wavelet bases defined in  Hammond et al (2011), where $g$ is required to meet some conditions, e.g $g$ is a band-pass filter. Obviously $g(s\lambda)=e^{s\lambda}$ can not meet these requirements.

Second, you call  $\hat{x}=\phi_s^{-1}x$ as graph wavelet transform, but actually it is not. As defined in  Hammond et al (2011), if $\phi_s$ is the graph wavelet bases, then $\hat{x}=<\phi_s, x>=\phi_s^*x$ is the graph wavelet transform.  Furthermore, I am more confusing about relationship between the graph wavelet transform and inverse graph wavelet transform defined in this paper (Eq.(4).).

Could you explain these for me?

---

> ### Public Comment · ~Jie_Zhang22 · 2025-03-15
> **The definition in this paper is not graph wavelet transform.**
>
> I agree with your view. The definition proposed in this paper is not a graph wavelet transform at all, but still a graph Fourier transform. First, graph wavelet transform requires the filter to be a band-pass filter, while GWNN uses the Heat Kernel, which is a low-pass filter. Second, graph wavelet transform is a discrete orthogonal transform, while $e^{-tL}$ is not a orthogonal matrix. I wrote a Chinese article to discuss this, see https://zhuanlan.zhihu.com/p/31156329132.

---

### Meta-Review · Area_Chair1 · 2018-12-14
**some novelty**

**Confidence:** 5
**Recommendation:** Accept (Poster)

**Metareview:**

AR1 and AR3 have found this paper interesting in terms of replacing the spectral operations in GCN by wavelet operations. However, AR4 was more critical about the poor complexity of the proposed method compared to approximations in Hammond et al. AR4 was also right to find the proposed work similar to Chebyshev approximations in ChebNet and to highlight that the proposed approach is only marginally better than GCN. On balance, all reviewers find some merit in this work thus AC advocates an accept. The authors are asked to keep the contents of the final draft as agreed with AR4 (and other reviewers) during rebuttal without making any further theoretical changes/brushing over various new claims/ideas unsolicited by the reviewers (otherwise such changes would require passing the draft again through reviewers).